# Behavioral, Physiological, and Pathological Approaches of Cortisol in Dogs

**DOI:** 10.3390/ani14233536

**Published:** 2024-12-07

**Authors:** Sorin Marian Mârza, Camelia Munteanu, Ionel Papuc, Lăcătuş Radu, Petraru Diana, Robert Cristian Purdoiu

**Affiliations:** 1Clinical Sciences Department, Faculty of Veterinary Medicine, University of Agricultural Sciences and Veterinary Medicine, 3-5 Manastur Street, 400372 Cluj-Napoca, Romania; sorin.marza@usamvcluj.ro (S.M.M.); ionel.papuc@usamvcluj.ro (I.P.); radu.lacatus@usamvcluj.ro (L.R.); diana.petraru@student.usamvcluj.ro (P.D.); robert.purdoiu@usamvcluj.ro (R.C.P.); 2Biology Section, Faculty of Agriculture, University of Agricultural Sciences and Veterinary Medicine, 3-5 Manastur Street, 400372 Cluj-Napoca, Romania

**Keywords:** cortisol, dogs, stress, Cushing’s disease, cortisol measurement, behavior, non-invasive methods

## Abstract

This review explores cortisol’s physiological and behavioral role in canines, focusing on its effects on stress, immune function, and metabolism. Abnormal cortisol levels’ clinical and behavioral consequences are discussed, with a particular focus on disorders like Cushing’s disease and behavioral problems brought on by stress. In addition to suggesting avenues for future study to enhance canine health and stress management, the findings highlight the significance of cortisol monitoring in veterinary practice and animal welfare.

## 1. Introduction

Cortisol is the predominant glucocorticoid produced by the adrenal glands in dogs, playing a central role in regulating metabolic and physiological responses. It is secreted in higher quantities than other glucocorticoids, such as corticosterone, and has significant biological potency. Cortisol’s effects span multiple systems, influencing glucose metabolism, immune response, and adaptation to stressors, thus impacting the body’s overall metabolic condition [1]. In both humans and animals, cortisol is involved in regulating the body’s metabolism, immune response, and overall ability to cope with physical and emotional stress [2]. For dogs, cortisol serves as a vital indicator of their health, particularly regarding how they respond to environmental stressors such as changes in routine, social interactions, and even long-term exposure to stressful environments like shelters [3]. Delving deeper into the mechanism, cortisol acts as a key messenger due to its fat-soluble nature, allowing it to cross the cell membrane [4]. Once inside the cell, it binds to specific receptors located in the cytoplasm. Upon binding, the Hsp90 protein dissociates from the receptor. In the absence of cortisol, the Hsp90 chaperone protein remains bound to the glucocorticoid receptor [5]. Once the cortisol–receptor complex forms, it translocates into the nucleus, where it influences gene transcription (Figure 1) [6].

The study of cortisol levels in dogs has gained significant attention in veterinary medicine and animal behavior research due to its direct relationship with health conditions such as Cushing’s disease and its potential to reveal underlying behavioral issues like anxiety and aggression. By understanding cortisol’s physiological role and the factors that influence its production and regulation, veterinarians and animal behaviorists can better diagnose and manage stress-related disorders in dogs. This review aims to provide a comprehensive exploration of the role of cortisol in canine health. In this review, ‘canine context’ refers to the specific living environments of dogs, such as pet homes, shelters, and competitive or working settings. Dogs are classified based on their current environment; for instance, shelter dogs that become pets are categorized as pet dogs once they are adopted into a home setting. This approach allows for a more accurate assessment of cortisol dynamics across different living conditions. Additionally, this review will explore the clinical and behavioral implications of abnormal cortisol levels and the importance of integrating cortisol monitoring into regular veterinary assessments.

Oxytocin and cortisol levels in both dog owners and their dogs are influenced by their behavioral interactions, with positive interactions leading to reduced cortisol levels in both parties [7]. The synchronization of stress markers, such as cortisol, between humans and dogs highlights the important role of emotional bonding in modulating stress responses (Figure 2) [8].

Restricted movement and stimulation-impoverished environments each exert distinct psychophysiological effects on dogs. Limitations in movement primarily influence the musculoskeletal and autonomic systems, often resulting in heightened stress due to spatial restriction and reduced physical activity. In contrast, environments lacking adequate sensory and cognitive stimulation mainly impact cognitive and sensory systems, leading to stress through insufficient engagement with the surroundings. Distinguishing these effects allows for a more nuanced understanding of the factors contributing to cortisol elevation and stress in canine populations [9]. Nocturnal activity and restless behavior were positively correlated with elevated cortisol concentrations in kenneled dogs, emphasizing the importance of providing adequate space and enrichment in housing conditions [10].

Non-invasive methods are particularly useful in non-clinical settings. For instance, Sechi et al. (2017) found that certain nutraceuticals, including L-theanine, alpha-casozepine, and tryptophan, have been studied. Stress can be categorized as acute or chronic, with acute stress responses typically providing short-term survival for their effects on reducing stress in dogs. L-theanine, an amino acid derived from green tea, promotes relaxation without causing drowsiness by modulating neurotransmitters associated with calming effects. Alpha-casozepine, derived from milk protein, has anxiolytic properties and has been shown to reduce behavioral signs of anxiety. Tryptophan, a precursor to serotonin, supports mood regulation and can help mitigate stress responses. Together, these nutraceuticals contribute to lower cortisol levels and improve stress resilience in dogs. [11].

Blood cortisol measurement remains a highly accurate method for evaluating stress and diagnosing endocrine disorders, although the invasive nature of blood sampling can trigger temporary stress-induced cortisol spikes [12]. Hair cortisol measurements, as discussed by Wirobski et al., offer a valuable long-term assessment tool, particularly useful for monitoring stress in working dogs and service animals [13]. Additionally, Wojtaś et al. [14] highlighted the use of hair cortisol levels to assess chronic stress in dogs housed in pet hotels, demonstrating its versatility across different settings. Fecal cortisol assays, as shown in studies by Wojtaś et al., can be applied in contexts such as search-and-rescue operations, where non-invasive sampling is crucial for monitoring canine health [14].

In competitive environments, cortisol levels can fluctuate based on the intensity of activities. Studies by Buttner et al. revealed that dogs participating in competitive sports exhibited synchronized cortisol fluctuations with their human handlers, indicating how both acute stress and social bonds affect cortisol regulation [15]. Similarly, studies by Morrow et al. demonstrated breed-dependent differences in cortisol responses during fear-related avoidance behaviors, suggesting that certain breeds may be predisposed to heightened cortisol reactions in stressful scenarios [16]. Cortisol is one of the primary glucocorticoids produced by the adrenal glands and plays a critical role in maintaining homeostasis in the body, especially in stress response. To provide a foundation for understanding cortisol’s role in canine physiology, it is essential to introduce the concept of stress. According to Donald Broom, stress is defined as a biological response elicited when an animal perceives a challenge or threat to its homeostasis, triggering adaptive physiological and behavioral changes aimed at coping with the stressor. Stress can be categorized as acute or chronic, with acute stress responses typically providing short-term survival benefits, while chronic stress can lead to detrimental effects on health and well-being. This framework is particularly relevant in veterinary science, as stress responses in animals are closely linked to welfare, behavior, and health outcomes [17].

## 2. Materials and Methods

This review focused on various observational studies involving dogs, assessing cortisol levels under different conditions such as shelter environments, pet ownership, and veterinary care. The studies reviewed included case reports and cross-sectional designs, as well as longitudinal studies analyzing chronic and acute stress in dogs. Both invasive and non-invasive sampling methods were considered for measuring cortisol levels across diverse canine contexts.

A thorough search of peer-reviewed literature was conducted using databases such as PubMed, Scopus, and Web of Science. Keywords including “cortisol in dogs”, “canine stress”, “salivary cortisol”, “hair cortisol”, “Cushing’s disease”, and “chronic stress in dogs” were employed to identify relevant studies published between 1999 and 2023. The references of selected papers were also examined to ensure no relevant studies were omitted. To be included in this review, studies had to meet specific inclusion criteria: they involved healthy dogs from various breeds, age groups, and sexes, and they examined the impact of environmental factors such as housing conditions, human interaction, and exposure to auditory stimuli (e.g., classical and rock music) on cortisol regulation. Only articles published in English were considered. Studies that reported unreliable or insufficient cortisol measurement data were excluded to maintain scientific rigor.

The studies reviewed employed a variety of cortisol measurement techniques, including salivary cortisol (non-invasive), blood cortisol (invasive), urine cortisol (minimally invasive), hair cortisol (chronic stress assessment), and fecal cortisol (long-term monitoring). For each study, details about the subjects, study design, and cortisol measurement methods were collected, along with information on environmental variables that could influence cortisol regulation. The selected studies were critically appraised to ensure they provided robust and valid cortisol assessments under various conditions. Special attention was given to studies that involved dogs in real-world scenarios, such as shelters or home environments, as these contexts provide practical insights into cortisol fluctuations.

Additionally, studies that required ethical approval included their ethical approval codes and the relevant granting body. Studies involving species other than dogs or those using unreliable cortisol measurement methods were excluded. The data from selected studies were critically evaluated based on their relevance to canine cortisol dynamics, and studies involving both acute and chronic stress were prioritized. The inclusion of a wide range of study designs allowed for a comprehensive understanding of cortisol regulation in dogs, which was essential for drawing meaningful conclusions about the effects of environmental factors on canine stress.

The original images were created using the Sketchbook iPadOs 18.1.1. application on an iPad Air (5th generation).

## 3. Cortisol Measurement Techniques

A range of cortisol measurement techniques were employed across the studies. Salivary cortisol was measured using a non-invasive collection of saliva samples, which were obtained through the use of cotton swabs. This method was favored for its ability to reflect real-time stress levels without significantly affecting the animal. Blood cortisol levels were measured using invasive blood sampling techniques, particularly in studies focused on clinical conditions such as Cushing’s disease. Although blood sampling is one of the most accurate methods, it may induce temporary increases in cortisol due to the stress of the procedure. The daily rhythm of glucocorticoids in dogs is governed by circadian patterns, with peak cortisol levels typically occurring in the morning and lower levels in the evening. However, this rhythm is influenced by environmental factors, individual variability, and acute stressors, which can disrupt the natural pattern. Acknowledging these fluctuations is crucial when interpreting cortisol data, as sampling time can significantly affect results [18]. Urine cortisol levels were measured by collecting urine samples to calculate the cortisol-to-creatinine ratio. This method is minimally invasive and is particularly useful for assessing chronic stress, although it may not be as effective for detecting acute cortisol fluctuations. Hair cortisol analysis was used in long-term studies to assess chronic stress. Hair samples were collected by cutting small sections near the root, and cortisol was extracted and analyzed using enzyme immunoassay (EIA) methods. Finally, fecal cortisol levels were measured by analyzing glucocorticoid metabolites excreted in feces, offering an indirect measure of cortisol production over time. This method was particularly useful for studying long-term cortisol fluctuations in non-invasive settings.

### Data Collection and Analysis

Cortisol levels were measured using enzyme-linked immunosorbent assay (ELISA) in most of the studies reviewed. The data were analyzed using statistical methods to identify significant differences in cortisol levels between different groups, such as shelter dogs versus pet dogs, or before and after specific interventions aimed at reducing stress.

## 4. Physiological Role of Cortisol in Dogs

Cortisol is synthesized in the adrenal cortex, specifically in the zona fasciculata, in response to stimulation by the hypothalamic–pituitary–adrenal (HPA) axis [19]. The HPA axis is triggered when the hypothalamus releases corticotropin-releasing hormone (CRH), which stimulates the pituitary gland to secrete adrenocorticotropic hormone (ACTH) [20]. ACTH then acts on the adrenal glands, prompting the release of cortisol into the bloodstream (Figure 3). Cortisol is essential to a dog’s ability to cope with stress, regulate metabolism, and modulate immune system responses. The regulation of cortisol in dogs is a complex process influenced by various interacting factors, including environmental conditions, physiological mechanisms, and social interactions. Environmental factors, such as housing conditions or exposure to stressful stimuli, directly impact cortisol levels by triggering the hypothalamic–pituitary–adrenal (HPA) axis, the primary pathway for cortisol release. Physiologically, cortisol secretion is regulated by an intricate feedback loop involving the hypothalamus, pituitary gland, and adrenal glands, allowing the body to adapt to acute and chronic stressors. Social factors, such as interactions with humans or other animals, also play a significant role; positive interactions are shown to reduce cortisol levels, while negative or unpredictable social environments can elevate them. Understanding these distinct yet interconnected influences gives us a clearer picture of how cortisol dynamics shape canine health and behavior.

Once released, cortisol circulates in the blood in two forms: bound to proteins like corticosteroid-binding globulin (CBG) or in its free, biologically active form. Free cortisol enters cells by crossing the cell membrane, binding to glucocorticoid receptors, and regulating gene expression within the nucleus (Figure 4) [21]. This mechanism helps the body manage inflammation, regulate glucose metabolism, and maintain overall energy balance during stress. Increases in cortisol levels are not necessarily bad but rather are necessary to better perform cognitive and emotional tasks such as those faced by a guide dog. Also, dogs belonging to those who regard their animals as social partners or meaningful companions have been shown to have relatively low salivary cortisol concentrations [22]. In ecological and evolutionary contexts, the ‘shy vs. bold’ behavioral syndrome describes consistent individual differences in behavior, where ‘bold’ animals tend to exhibit exploratory and risk-taking behaviors, while ‘shy’ animals display more cautious and risk-averse tendencies. Research suggests that these behavioral syndromes are associated with differential glucocorticoid (GC) reactivity. Bold individuals often show a lower baseline but a more rapid increase in GC levels during acute stress, facilitating quick mobilization of resources in novel or challenging situations. In contrast, shy individuals may exhibit higher baseline GC levels, reflecting a heightened sensitivity to environmental stressors. This variability in GC reactions supports adaptive strategies for different ecological niches, with shy and bold behavioral types each benefiting from unique physiological stress profiles that influence health, survival, and fitness. Studies by Horvath et al. have explored the complex relationship between play, displacement behavior, and glucocorticoid (GC) levels in working dogs, particularly police dogs. Their findings suggest that play can serve as a mechanism to alleviate stress and reduce GC levels, promoting a balanced stress response in high-stress environments. However, ambivalent or displacement behaviors, which are sometimes observed in these dogs under conflicting motivations or stress, have been correlated with elevated GC levels, indicating underlying tension and physiological arousal. These behaviors, observed during both work and rest periods, highlight the need for structured play and positive outlets for stress in working dogs [23]. A growing body of research has examined the effects of aversive training methods, such as shock collars, leash corrections, and other punishment-based approaches, on glucocorticoid (GC) responses in dogs. Studies indicate that aversive methods are associated with elevated GC levels, reflecting heightened stress and anxiety. Reviews have shown that these training techniques not only increase physiological stress markers but also contribute to behavioral issues, such as increased aggression and fearfulness, which can further elevate GC responses. The negative impact of aversive methods underscores the importance of adopting positive reinforcement-based techniques, which have been shown to promote learning and reduce stress without triggering excessive GC secretion [24,25].

### Acute vs. Chronic Cortisol Elevation

Recent studies have further expanded our understanding of how cortisol regulates key physiological functions in dogs, including immune modulation and energy balance. According to research, chronic stress leads to immune suppression, increasing a dog’s susceptibility to infections and illnesses [26]. The regulation of cortisol secretion through the hypothalamic–pituitary–adrenal (HPA) axis follows a precise sequence. In response to stress, the hypothalamus releases corticotropin-releasing hormone (CRH), which stimulates the anterior pituitary gland to secrete adrenocorticotropic hormone (ACTH). ACTH then travels through the bloodstream to the adrenal cortex’s zona fasciculata, prompting cortisol synthesis and release. Circulating cortisol exerts physiological effects on metabolism, immune function, and stress adaptation. Elevated cortisol levels subsequently provide negative feedback to the hypothalamus and pituitary, modulating further CRH and ACTH release to maintain hormonal balance [27].

Cortisol levels in dogs can vary depending on whether the stressor is acute or chronic. In acute stress, cortisol levels rise rapidly, allowing the body to mobilize energy reserves, suppress non-essential processes like digestion, and heighten focus on immediate threats [28]. For instance, a sudden loud noise or an unfamiliar environment may trigger this response. While this temporary spike in cortisol is necessary for survival, chronic elevation can have detrimental effects [27].

Chronic stress, which results in prolonged cortisol elevation, can lead to negative health outcomes such as immunosuppression, muscle breakdown, and weight gain [4]. In dogs, chronic cortisol elevation is often associated with conditions like Cushing’s disease, where excessive production of cortisol leads to symptoms such as increased thirst, excessive eating, weight gain, and muscle weakness [29]. Chronic stress has also been linked to behavioral changes in dogs, including heightened anxiety, aggression, and a diminished ability to cope with environmental changes [30]. Understanding the balance between acute and chronic cortisol levels is vital in managing both the health and behavior of dogs.

## 5. Methods of Measuring Cortisol Levels in Dogs

Accurate measurement of cortisol levels in dogs is crucial for assessing their stress response, overall health, and behavioral conditions. Each method has specific advantages, and the choice of approach depends on the context in which the measurement is taken, such as assessing acute or chronic stress.

Accurate measurement of cortisol levels in dogs is crucial for assessing their stress response, overall health, and behavioral conditions. Various methods exist, ranging from invasive techniques such as blood, saliva, and urine testing, to non-invasive alternatives like hair and fecal cortisol assays (Bowman et al.) [31]. Each method has specific advantages, and the choice of approach depends on the context in which the measurement is taken, such as assessing acute or chronic stress. The comparison of cortisol measurement methods shows that blood sampling is highly accurate but invasive, making it ideal for clinical diagnostics despite potentially causing stress [32]. Saliva sampling offers a balance between moderate invasiveness and high accuracy, making it suitable for real-time stress monitoring. Urine sampling provides moderate accuracy for chronic cortisol assessment but requires close monitoring for sample collection. Hair and fecal sampling are minimally invasive and provide moderate accuracy, making them ideal for long-term, non-invasive monitoring of chronic stress [33]. Overall, the method chosen should balance the need for accuracy with the importance of minimizing stress in the dog.

### 5.1. Invasive Methods: Blood, Saliva, and Urine Testing

Blood sampling remains one of the most direct methods for measuring cortisol in dogs. It provides an immediate measure of circulating cortisol, particularly valuable for diagnosing endocrine disorders like Cushing’s disease. However, the procedure can induce stress, potentially causing a temporary rise in cortisol levels and altering the results. Despite this, studies such as those by Rosado et al. have shown the effectiveness of blood cortisol measurements in understanding physiological stress in dogs [34].

Salivary cortisol testing offers a less invasive alternative, reflecting free cortisol levels that are not bound to proteins, making it a more accurate measure of the biologically active hormone. M.L. Cobb et al. conducted a meta-analysis on salivary cortisol measurements and concluded that it is a reliable tool for assessing stress in dogs [35]. The impact of invasive procedures, such as blood sampling, on cortisol levels in dogs has been widely studied, and methods to mitigate stress following these procedures are of great interest. Hennessy et al. demonstrated that gentle massage following blood sampling can reduce cortisol levels, indicating a calming effect that helps mitigate the stress response. This technique highlights the potential of post-procedural soothing interventions to enhance welfare during veterinary procedures, supporting a reduction in stress-related cortisol elevation in dogs [36]. This method has been applied in studies on the impact of human interaction on cortisol levels in shelter dogs, showing its effectiveness in monitoring real-time changes in stress levels. Salivary cortisol is increasingly used as a non-invasive biomarker for assessing acute stress in dogs. Its reliability has been confirmed in various stress-inducing situations such as veterinary visits and separation from owners, making it a practical tool for studying stress physiology. The correlation between stress-inducing stimuli and salivary cortisol levels has been well-documented, emphasizing its potential in evaluating not only acute but also chronic stress in canines [37].

The urine cortisol-to-creatinine ratio is another method used in veterinary settings to assess cortisol levels, particularly for diagnosing chronic endocrine disorders such as hypercortisolism. Studies by Quilez et al. have highlighted the utility of urine testing in diagnosing hypercortisolism and monitoring the effects of treatments [38]. However, urine testing may not reflect acute stress due to the lag in cortisol excretion, and its use is often restricted to clinical cases rather than behavioral studies.

### 5.2. Non-Invasive Methods: Hair and Fecal Cortisol Assays

Hair cortisol measurement has gained recognition as an effective tool for monitoring long-term cortisol accumulation, offering a retrospective insight into chronic stress. As cortisol is incorporated into the hair shaft during growth, it provides a cumulative record of the dog’s stress levels over time. Grigg et al. demonstrated the utility of hair cortisol in assessing chronic stress, particularly in shelter environments, where long-term stress can significantly affect a dog’s behavior and health [39].

Fecal cortisol assays provide another non-invasive method, measuring glucocorticoid metabolites excreted in feces. This method reflects cortisol production over time, allowing for the assessment of chronic stress without the need for invasive procedures. Accorsi et al. showed that fecal cortisol is a reliable marker for evaluating the effects of training and environmental changes on stress levels in dogs [40]. Fecal sampling has been particularly useful in shelter settings, where repeated handling for invasive measures may exacerbate stress.

Hair cortisol concentration offers a promising method for assessing long-term stress in dogs, with this study highlighting its sensitivity to both internal factors, such as age, and external stressors like environmental changes. The complexity of hair cortisol as a biomarker is evidenced by its responsiveness to various stressors, emphasizing the need for a multifactorial approach when interpreting results.

## 6. Cortisol Levels in Various Canine Contexts

Cortisol levels in dogs can vary significantly depending on their environment, experiences, and interactions with humans and other animals. Research on cortisol dynamics has explored various contexts in which cortisol levels fluctuate, such as shelter dogs versus pet dogs, the effects of human interaction, and the influence of environmental factors like music and training. Beerda et al. conducted several foundational studies on stress and glucocorticoid levels in dogs, exploring the physiological and behavioral impacts of stress-inducing stimuli, such as shock collars and other aversive methods. Their findings highlighted a significant increase in cortisol levels following the use of shock collars, indicative of stress and potential welfare implications. Similarly, Salgiri et al. demonstrated comparable results, emphasizing the importance of avoiding aversive techniques that elicit prolonged physiological stress in dogs [3,41].

### 6.1. Cortisol in Shelter Dogs vs. Pet Dogs

The environment in which a dog is housed plays a significant role in determining its cortisol levels. Shelter dogs, particularly those living in high-stress environments like kennels, tend to exhibit elevated cortisol levels compared to pet dogs living in stable home environments. Studies by Part et al. found that the physiological, physical, and behavioral changes associated with kenneling led to heightened stress levels, as indicated by increased cortisol concentrations [42]. This finding is further supported by those who documented that dogs in kennels often show elevated salivary cortisol levels in response to the chronic stress of confinement [43].

Dogs with consistent routines and positive interactions generally exhibit lower cortisol levels, as these factors provide stability and reduce stress. Consistent routines refer to a predictable daily structure, including regular feeding times, exercise schedules, and rest periods, which help reduce uncertainty and create a stable environment for the dog. Positive interactions involve engaging, non-threatening social interactions with humans or other animals, such as petting, play, training, and gentle handling, which reinforce trust and emotional security. Together, these conditions contribute to a balanced stress response and support lower baseline cortisol levels. Additionally, Hewison et al. found that preventing visitor access to kennels reduced noise and alleviated some behavioral and physiological stress responses, including lower cortisol levels, suggesting that kennel environments often exacerbate stress [44].

Human interaction plays a key role in moderating these cortisol spikes. Shiverdecker et al. (2013) discovered that shelter dogs that experienced more human interaction exhibited lower cortisol levels and reduced stress-related behaviors, indicating that consistent positive interaction can buffer against the negative effects of confinement [45].

Similarly, Bergamasco et al. (2010) showed that heart rate variability and salivary cortisol levels decreased after positive human–dog interactions, further supporting the importance of social engagement in mitigating stress [46].

Separation from familiar humans is a significant source of stress for dogs, triggering physiological and behavioral responses indicative of elevated cortisol levels [47]. Research by Rehn et al. has shown that dogs experience increased stress levels during periods of separation from their owners, as indicated by behavioral signs of distress and elevated cortisol measurements. These findings underscore the importance of secure human–dog bonds in mitigating separation-related stress and suggest that attachment quality may influence a dog’s resilience to separation [48]. The housing setup in shelters also plays a crucial role. Villa et al. (2012) compared group and pair housing conditions and found that dogs in group housing exhibited lower cortisol levels and fewer stress-related behaviors, such as excessive barking and pacing [47]. This suggests that intraspecific interactions (i.e., interactions with other dogs) in group housing help alleviate stress. Alberghina et al. (2019) supported these findings by observing that dogs housed together exhibited lower daily fluctuations in cortisol compared to isolated dogs, indicating the value of social exposure in reducing stress [49].

Despite the benefits of social interaction and human engagement, chronic exposure to stressful conditions in kennels can lead to persistent behavioral issues. Denham et al. (2014) highlighted the occurrence of repetitive behaviors in kenneled dogs, which are often linked to prolonged elevated cortisol levels, suggesting that these behaviors may be a coping mechanism for chronic stress [50]. Buttner et al. (2022) also noted that extreme life histories, such as long-term confinement in shelters, are associated with higher cortisol levels and altered social behaviors, indicating the long-term effects of shelter stress on dogs’ endocrine responses [51]. Conversely, pet dogs, which typically enjoy consistent routines, positive interactions with their owners, and enriched environments, often show more stable cortisol levels. Wirobski et al. (2021) extended this finding by comparing cortisol and oxytocin levels in pet dogs and pack-living dogs, revealing that domesticated dogs have adapted to reduced stress through long-term interaction with humans [13].

Interestingly, certain studies have shown that shelter dogs can adapt to new environments over time. Wojtaś et al. (2022) reported that dogs staying in a pet hotel exhibited high cortisol levels during initial confinement, but these levels gradually decreased as the dogs adapted to their new surroundings [14].

This indicates that with the right environmental enrichment and social interaction, stress levels in kenneled dogs can be reduced, though the adaptation period varies based on individual dogs and the level of enrichment provided.

Finally, d’Angelo et al. (2021) explored how animal-assisted interventions in a prison setting affected cortisol levels in shelter dogs [52]. They found that positive engagement in structured activities led to reduced stress markers, providing another example of how social and environmental factors can influence cortisol levels in dogs.

### 6.2. The Effect of Human Interaction on Cortisol Levels

Psychobiological factors, such as the relationship quality between humans and their dogs, significantly influence cortisol variability in human–dog dyads [53]. Positive interactions, such as petting, playing, and simply being in the presence of familiar humans, have been shown to reduce stress in dogs. A study by Shiverdecker et al. demonstrated that shelter dogs experienced significant reductions in plasma cortisol levels following interaction with humans [45]. This finding suggests that regular human contact can mitigate the stress associated with confinement and social isolation. A study revealed that interactions between dog owners and their dogs lead to an increase in oxytocin levels in both parties, while cortisol levels decrease in the owners but show an upward trend in the dogs [53].

The Bowlby–Ainsworth concept of ‘safe-haven’ and ‘secure base’ is central to attachment theory, describing how bonded individuals, such as caregivers or familiar companies, provide comfort (safe haven) and a secure foundation (secure base) from which animals can explore and engage with their environment. In dogs, access to a trusted human or companion animal can act as a stress buffer, reducing anxiety and promoting adaptive behaviors. This attachment bond helps dogs regulate their stress responses, as the presence of a ‘safe-haven’ lowers cortisol levels, reinforcing emotional security and enhancing overall welfare.

Similarly, studies like that of Handlin et al. found that short-term interactions between dogs and their owners resulted in lower cortisol levels in dogs, indicating the calming effect of owner–dog relationships [54]. These interactions can serve as a form of social buffering, helping to reduce the physiological stress response in dogs. Considering both dog and human personalities when pairing dogs with people can help minimize behavioral conflicts within the dog–human relationship by avoiding mismatches. Notably, significant correlations have been found between the personality traits of openness and agreeableness and the owner’s satisfaction with the dog–human bond [55].

The presence of dog owners during veterinary examinations has been found to significantly alleviate both behavioral and physiological signs of stress, such as reduced heart rate and lower salivary cortisol levels, highlighting the calming effect of human–animal interactions. These results emphasize the importance of human presence in stress management for dogs, as interactions with their owners can positively influence stress markers, suggesting a meaningful impact on their overall well-being during stressful situations [56].

Also, owner presence during veterinary examinations has been shown to significantly reduce both behavioral signs of stress and physiological markers such as heart rate and salivary cortisol, suggesting that owner–dog interaction plays a crucial role in mitigating stress responses [57].

Additionally, social exposure among dogs led to a notable interaction effect, influencing the cortisol-to-creatinine ratio (C/Cr), and highlighting the importance of social factors in modulating physiological stress responses [52].

Recent studies have expanded this understanding of human–dog interactions. For instance, Menor-Campos et al. (2011) emphasized the importance of regular exercise and human interaction in reducing cortisol levels in shelter dogs [58]. Their study found that dogs who had more frequent contact with humans exhibited lower cortisol levels, reinforcing the idea that positive human interaction can buffer stress. Furthermore, the influence of owner behavior extends beyond familiar human–dog bonds. According to the research by Pinelli et al. (2023), even unfamiliar human interactions can have a buffering effect on cortisol levels in kennel-residing dogs [59]. In a study by Shin and Shin (2017), the researchers looked at the relationship between sociability toward humans and physiological stress in dogs [60]. Their results demonstrated that dogs who displayed more sociable behavior toward humans had lower cortisol levels compared to less sociable dogs, which further emphasizes the link between human interaction and stress modulation in canines.

Lastly, Schöberl et al. (2017) examined psychobiological factors in human–dog dyads, showing that strong emotional bonds help regulate cortisol levels. Dogs with stronger attachments to their human companions had more stable cortisol levels, highlighting the stress-reducing power of positive human–dog relationships [61].

### 6.3. Influence of Music, Training, and Other Environmental Factors on Cortisol Levels

In addition to human interaction, environmental factors such as music and training have been shown to influence cortisol levels in dogs. Research by Bowman et al. [31] explored the effect of different genres of music on the stress levels of kenneled dogs, finding that certain types of music, particularly classical, were associated with lower cortisol levels. This suggests that environmental enrichment strategies, such as playing calming music in kennels, may help reduce stress and improve the welfare of shelter dogs.

Training activities can also have a significant impact on cortisol levels. While physically and mentally engaging dogs, training can either increase or decrease cortisol levels depending on the methods used and the dog’s reaction to the training environment. Studies have examined the effects of different cognitive and physical activities on salivary cortisol levels in dogs, showing that structured, positive training experiences can reduce stress [62,63].

Other factors, such as changes in housing conditions, noise levels, and social interactions with other dogs, can also influence cortisol levels. In particular, studies have shown that intraspecific interactions, or interactions between dogs, can either increase or decrease stress depending on the dynamics between the animals [64]. The careful management of these environmental factors can help optimize cortisol regulation and overall well-being in dogs [65].

Franzini de Souza et al. (2018) demonstrated that dogs with sound sensitivity often show heightened cortisol levels in response to loud or sudden noises, further emphasizing the need for noise control in kennel environments to manage stress [33]. Bowman et al. (2017) found that genres like classical music significantly reduced stress, while heavy metal increased cortisol levels in kenneled dogs [31]. Managing auditory stimuli, therefore, plays a crucial role in regulating cortisol. Wormald et al. (2016) highlighted that inter-dog aggression in social settings elevates cortisol, suggesting that careful management of dog–dog interactions is essential for stress reduction [66]. Implementing both social and environmental enrichment strategies can balance cortisol regulation effectively.

## 7. Clinical and Behavioral Implications of Abnormal Cortisol Levels

Cortisol plays a critical role in maintaining the physiological and psychological balance in dogs, and abnormalities in cortisol levels can have profound clinical and behavioral implications [67]. Both hypercortisolism (elevated cortisol levels) and hypocortisolism (low cortisol levels) are associated with various health disorders and behavioral changes in dogs, highlighting the importance of monitoring cortisol levels in veterinary practice. Excess cortisol production in hypercortisolism can result in significant health issues, including hypertension with damage to vital organs, an increased risk of clotting and thrombosis, and proteinuria [68]. Additionally, it complicates the treatment of other conditions, such as diabetes mellitus, by worsening the metabolic control of these comorbidities [69]. The MDR1 (multidrug resistance 1) defect, a genetic mutation affecting P-glycoprotein function, has implications for cortisol metabolism and drug sensitivity in certain breeds, such as Collies and Australian Shepherds. This mutation can influence adrenal function and stress responses, necessitating careful consideration during clinical evaluations. A key gatekeeper at the blood–brain barrier (BBB), the multidrug-resistant gene 1-type p-glycoprotein (MDR1 p-gp), shields the central nervous system from the buildup of harmful medications and xenobiotics. Furthermore, MDR1 p-gp regulates the intracerebral access of glucocorticoids, affecting the hypothalamic–pituitary–adrenocortical (HPA) system’s ability to function. Thus, MDR1 p-gp plays a significant role in the brain’s exposure to and sensitivity to glucocorticoids by actively transporting corticosteroid hormones across the blood–brain barrier and out of the brain. Since steroid hormones control the expression of the murine mdr1b gene in a regulatory feedback loop and the modulation of intracellular glucocorticoid levels by MDR1 p-gp modifies GR translocation and signaling, there is a reciprocal interaction between the corticosteroid and MDR1 p-gp systems [70].

Research indicated that a mild increase in plasma corticosterone, induced by stress, was significantly reduced when the effects of acute MDR1 P-glycoprotein (p-gp) inhibition on the glucocorticoid system were assessed. In various studies, MDR1 p-glycoprotein inhibitors did not enhance the anti-anxiety effects of corticosterone, even though corticosterone exhibited an anxiolytic-like significant impact.

### 7.1. Hypercortisolism: Cushing’s Disease and Related Conditions

One of the most common disorders associated with elevated cortisol levels in dogs is Cushing’s disease, also known as hyperadrenocorticism. This condition is characterized by the excessive production of cortisol, often due to a pituitary or adrenal gland tumor [69]. Dogs with Cushing’s disease typically exhibit symptoms such as excessive thirst (polydipsia), increased appetite (polyphagia), weight gain, muscle weakness, and hair loss [71]. Over time, chronic exposure to elevated cortisol can lead to severe health complications, including diabetes mellitus, hypertension, and increased susceptibility to infections [72]. Thyroid dysfunctions, such as hypothyroidism, can alter glucocorticoid dynamics by affecting metabolism and the hypothalamic–pituitary–adrenal (HPA) axis. Thyroidal disturbances are often associated with behavioral changes, such as increased anxiety or lethargy, complicating the assessment of cortisol levels.

Research by Quilez et al. has highlighted the importance of urinary cortisol-to-creatinine ratios in diagnosing hypercortisolism, offering veterinarians a reliable tool to assess chronic cortisol elevation [38]. Additionally, case studies demonstrated that behavioral changes, such as increased irritability and aggression, may also accompany clinical signs of hypercortisolism, further complicating the condition [73]. Early detection through cortisol measurement is essential in managing this condition, as treatment options, including medication or surgery, can significantly improve the dog’s quality of life [74].

### 7.2. Behavioral Implications: Aggression, Anxiety, and Fear

Abnormal cortisol levels, whether elevated or suppressed, can also contribute to a range of behavioral issues in dogs. Studies have shown that chronic stress, resulting in sustained high cortisol levels, is linked to anxiety, aggression, and fear-related behaviors. Research by Rosado et al. found that dogs with aggressive tendencies had higher baseline cortisol levels, suggesting a correlation between stress and aggression [34]. Similarly, dogs with anxiety disorders, particularly separation anxiety, often exhibit elevated cortisol levels during episodes of stress, as highlighted in studies by Wojtaś et al. [14].

Fear-related behaviors, such as avoidance or hyperactivity, can also be exacerbated by cortisol dysregulation. Mary Morrow et al. explored breed-specific differences in cortisol responses to fear-inducing stimuli and noted that certain breeds may be more susceptible to fear-related cortisol spikes [16]. These findings underscore the need for individualized behavioral interventions, particularly in working breeds or those prone to anxiety and aggression.

Elevated cortisol concentrations were associated with fearfulness in juvenile dogs, with higher cortisol levels corresponding to difficulties in settling within unfamiliar environments [75]. Additionally, dogs with stronger motor lateralization, particularly those showing a marked preference for one paw, exhibited more confident and relaxed behavior when exposed to novel stimuli [76].

These conditions can lead to an exacerbation of stress-related behaviors through several interconnected mechanisms. Chronic stress activates the hypothalamic–pituitary–adrenal (HPA) axis persistently, resulting in prolonged cortisol release, which can impair immune function and disrupt normal physiological processes. Additionally, stress can heighten the animal’s sensitivity to environmental stimuli, lower its threshold for stress responses, and increase behaviors such as aggression, anxiety, and hypervigilance. These factors collectively exacerbate the intensity and frequency of stress-related behaviors over time. Garbiec et al. (2024) found that dogs with strong motor lateralization exhibited higher cortisol levels under stress, leading to increased anxiety and fear, indicating that physiological traits like lateralization impact stress resilience [77]. This intricate three-way interaction between cortisol levels, environmental conditions, and canine behavior highlights the importance of comprehensive cortisol monitoring as a tool for assessing canine welfare. Environmental stressors and social dynamics interact to influence cortisol regulation, which in turn affects both physiological responses and behavioral outcomes in dogs. Understanding these interconnections allows for more targeted interventions, ultimately enhancing canine health, welfare, and the effectiveness of stress management practices. Vékony and Pongracz (2024) highlighted that the higher-ranking dog in multi-dog households experienced cortisol spikes during competitive interactions, linking dominance to elevated stress and aggression in social hierarchies [78].

Routine practices, such as grooming, can also trigger stress. Jeong and Kim (2023) reported that clipper grooming raised cortisol levels and heightened anxiety in companion dogs, underlining the importance of managing stress even in routine care [79]. Additionally, Flint et al. (2024) demonstrated that daily cannabidiol (CBD) reduced cortisol and stress-related behaviors in dogs exposed to repeated car travel, suggesting potential therapeutic options for stress management [80].

Finally, Svobodová et al. (2014) revealed that prolonged cortisol elevation weakens immune function in dogs, increasing susceptibility to stress-related illnesses, and reinforcing the need for effective long-term stress management strategies [81].

### 7.3. Cortisol as a Biomarker in Veterinary Medicine

The use of cortisol as a biomarker for both clinical and behavioral issues has become increasingly common in veterinary medicine. Regular cortisol monitoring can aid in diagnosing endocrine disorders like Cushing’s disease and Addison’s disease (hypoadrenocorticism), where insufficient cortisol production leads to symptoms like lethargy, vomiting, and weight loss. In behavioral practice, cortisol measurements can help veterinarians and animal behaviorists assess the effectiveness of interventions designed to reduce stress, such as environmental enrichment, training modifications, or the introduction of calming therapies.

Research on the welfare effects of various training methods suggests that measuring cortisol levels can be an effective way to assess stress in dogs during different types of training. Similarly, research has demonstrated the role of cortisol in assessing the impact of shelter housing and the effectiveness of human interaction in reducing stress [82,83].

### 7.4. Cortisol and Cancer in Dogs

Generally, stress has repercussions on the whole organism which can be seen in the immunological and inflammatory paths. The general question is how? Cortisol released during stress can affect the rate of bacterial or viral infections, creating a favorable environment for their manifestation. It can also produce negative effects on the DNA structure. Many of the cancer-initiating mechanisms are affected by stress. Until now, there has been a debate that mental stress in animals can be linked to the occurrence of cancer. However, due to evolutionary, epidemiological, and physiological considerations, this link is difficult to prove [84]. States of fear, anxiety, and aggression are correlated with an increased incidence of cancer in dogs. The study that demonstrates this had 350 purebred and mixed-breed dogs [85]. Among the most common cancers that occur in dogs as a result of stress are osteosarcoma and lymphoma. Regarding osteosarcoma, there may be a connection between dogs’ large body size and the development of this type of cancer. Moreover, the involvement of different mechanisms in the occurrence of cancer does not exclude the strong link between stress and certain glucocorticoid (cortisol), adrenergic, immunological, and inflammatory signaling pathways [86,87]. The endocrine signaling mechanism includes sympathetic nervous system (SNS) activation, namely the release of catecholamines, and the hypothalamic–pituitary axis, with the release of cortisol [88,89]. Moreover, SNS signaling can also occur through vegetative nerves. The adrenergic pathway can regulate the effects of catecholamines by binding β2-adrenergic receptors and activating adenylyl cyclase, which converts ATP to cAMP. cAMP stimulates two major effectors, exchange factors directly regulated by cAMP (EPAC) and protein kinase A (PKA), which are then responsible for the activation of CREB and AP-1. The latter are transcription factors that modulate gene expression.

Cortisol initiates the stress pathway controlled by glucocorticoid receptors (GRs) [90]. Mechanisms modulated by stress are represented by essential molecular activities, such as proliferation, cell cycle regulation, DNA damage, and repair, which contribute to the initiation of cancer in case of dysregulation. Once translocated into the cytoplasm, glucocorticoids induce ROS/RNS release and activate the DNA damage response (DDR). In this signaling path, two enzymes have a major role, ataxia telangiectasia mutant (ATM) and ataxia telangiectasia related to Rad3 (ATR). They are mainly involved in responses to DDR. Stimulation of ATM and ATR results in phosphorylation of effector kinases, facilitating cell cycle arrest by inhibiting cyclin-dependent kinase, with particular importance in DNA damage repair and apoptosis initiation, by phosphorylating substrates such as p53 and BRCA1 [91]. In brief, it can be stated that there is a direct proportional link between cortisol, psychological stress, and cancer in dogs.

## 8. Conclusions and Future Perspectives

Cortisol is a vital biomarker for assessing both physiological and psychological health in dogs. Its role in regulating stress responses, metabolism, and immune function makes it a key indicator of well-being. As this review has demonstrated, cortisol levels can vary significantly depending on environmental factors, human interaction, and individual health conditions. Elevated cortisol levels, whether from chronic stress or clinical conditions like Cushing’s disease, can lead to severe health and behavioral issues, emphasizing the need for regular cortisol monitoring.

Methods of measuring cortisol in dogs have evolved to include both invasive and non-invasive techniques. While blood, saliva, and urine tests remain standard in clinical settings, non-invasive methods like hair and fecal cortisol assays offer valuable insights into long-term stress without adding to the dog’s burden. These methods are especially useful in shelter environments, where stress management is crucial for dog welfare.

The clinical and behavioral implications of abnormal cortisol levels highlight the importance of integrating cortisol measurements into routine veterinary practice. Regular monitoring can help detect endocrine disorders early and guide the management of stress-related behavioral problems, such as anxiety and aggression.

Future research should continue to explore the long-term effects of chronic cortisol elevation in dogs, particularly about environmental factors such as housing conditions and human interaction. Additionally, further investigation into the effectiveness of non-invasive cortisol measurement methods in various settings can provide more practical solutions for monitoring canine welfare.

Incorporating cortisol monitoring into standard veterinary assessments will not only enhance the ability to diagnose and manage stress-related conditions but also improve the overall quality of life for dogs. By better understanding how cortisol levels reflect health and behavior, veterinarians and researchers can develop more targeted strategies for promoting canine well-being.

Moreover, advancements in cortisol measurement techniques have provided new insights into the benefits of non-invasive methods.

Further research has shown that cortisol dysregulation is closely linked to behavioral disorders in dogs. For instance, dogs suffering from anxiety-related behaviors are often found to have elevated baseline cortisol levels. Studies by Handlin et al. showed that anxious dogs tend to exhibit higher salivary cortisol levels, particularly during separation from their owners [51].

Research by Sapolsky et al. [1] expands upon the cortisol release response, suggesting a multifaceted regulatory framework that involves permissive, suppressive, stimulatory, and preparative actions across different physiological systems. Rather than a simple reaction to stress, cortisol release is mediated by complex feedback mechanisms that integrate both immediate stress signals and longer-term environmental contexts. In dogs, this implies that cortisol levels are not merely reactive but are part of an adaptive system modulating various processes such as immune function, metabolism, and behavior to optimize survival and adaptability under varying conditions (Sapolsky et al. [1]). Understanding cortisol release within this broader framework can enhance our interpretation of cortisol dynamics and their implications for canine health and welfare [1]. The interaction between cortisol and sex steroids, including the effects of castration, plays a significant role in stress physiology. Castration has been shown to influence cortisol levels, potentially due to altered hormonal balance and stress response mechanisms. Understanding these interactions is critical for assessing the impact of reproductive status on cortisol dynamics in dogs [92].

The importance of cortisol monitoring in veterinary practice cannot be overstated. As new research continues to uncover the links between cortisol and canine health, integrating both invasive and non-invasive cortisol measurement techniques into routine veterinary care will be essential. Future studies should explore the long-term effects of interventions aimed at reducing chronic cortisol elevation, such as environmental enrichment and behavior modification programs. By deepening our understanding of cortisol dynamics in dogs, we can improve welfare standards and enhance treatment strategies for stress-related conditions in dogs.

## Figures and Tables

**Figure 1 animals-14-03536-f001:**
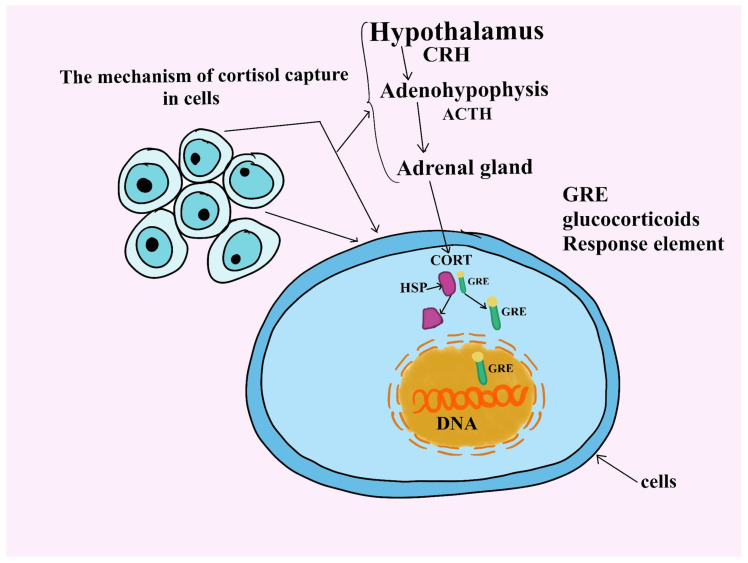
The mechanism of cortisol capture in cells. Once within the cell, it attaches itself to particular cytoplasmic receptors. The HSP protein separates from the receptor after binding. The HSP chaperone protein is still attached to the glucocorticoid receptor when cortisol is not present. The cortisol–receptor complex translocates into the nucleus after it has formed, affecting gene transcription.

**Figure 2 animals-14-03536-f002:**
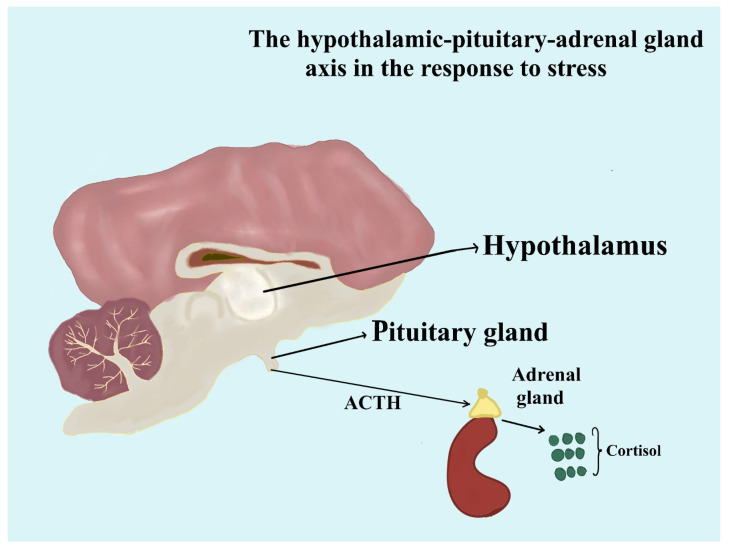
The hypothalamic–pituitary–adrenal gland axis is in response to stress. Under the influence of the hypothalamic–pituitary–adrenal (HPA) axis, the adrenal cortex, particularly the zona fasciculata, produces cortisol. Adrenocorticotropic hormone (ACTH) is secreted by the pituitary gland in response to stimulation by corticotropin-releasing hormone (CRH), which is released by the hypothalamus. Next, cortisol is released into the bloodstream by the adrenal glands as a result of ACTH’s action.

**Figure 3 animals-14-03536-f003:**
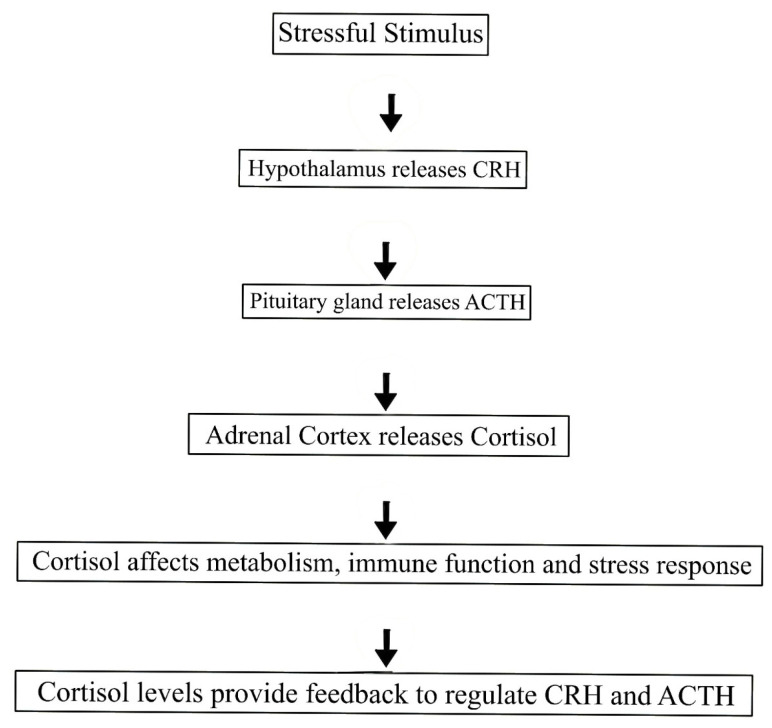
Cortisol regulation through the HPA axis illustrates how cortisol is produced and regulated in response to stress in dogs. The process starts with a stressful stimulus, which triggers the hypothalamus to release CRH (corticotropin-releasing hormone). CRH then signals the pituitary gland to release ACTH (adrenocorticotropic hormone), which travels through the bloodstream to the adrenal cortex. In response, the adrenal cortex releases cortisol, a hormone that helps regulate metabolism, immune function, and the body’s stress response. Once cortisol levels rise, they provide feedback to the hypothalamus and pituitary gland to reduce the release of CRH and ACTH, helping to regulate and balance cortisol production.

**Figure 4 animals-14-03536-f004:**
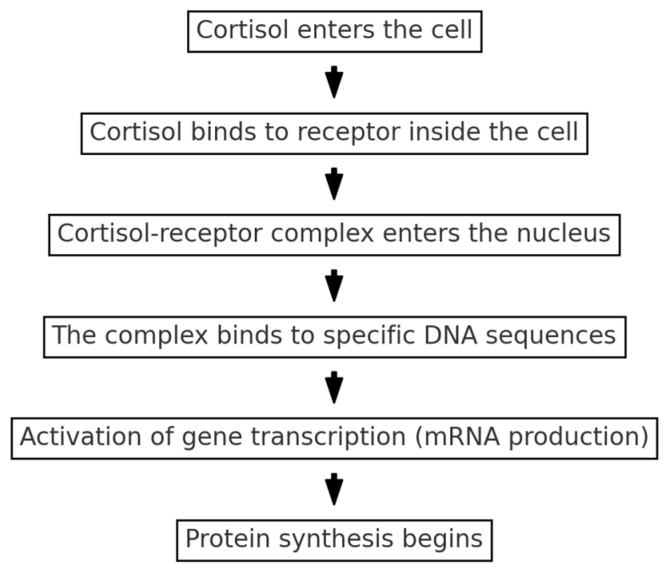
The figure illustrates the process by which cortisol interacts with a cell to regulate gene expression. Cortisol first enters the cell by diffusing through the plasma membrane and binds to the intracellular glucocorticoid receptor (GR), forming a complex. This cortisol–receptor complex then translocates into the nucleus, where it binds to specific DNA sequences known as glucocorticoid response elements (GREs). This binding activates gene transcription, leading to the production of mRNA. The mRNA is subsequently translated into proteins, which mediate cortisol’s effects on cellular functions such as metabolism and stress response.

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
