# Peer review of "Behavioral, Physiological, and Pathological Approaches of Cortisol in Dogs"

_animals, 2024, doi:10.3390/ani14233536_

Round 1
Reviewer 1 Report
Comments and Suggestions for Authors
General comment:
This contribution regards an issue of paramount interest in dog pathophysiology, id est the use (and misuse?) of cortisol levels as an indicator of mammalian stress. The topic is far for being new or original, yet the Authors attempt to exploit it in terms of sound indicator of psychophysical and immunological welfare in both pet and non-pet dogs. The part comparing circulating vs other source (feces, hair, etc..) is definitely of general interest for a variety of readers.
The major weakness concerns the total absence of the consideration of condition of EUSTRESS: some “limited” cortisol release is in fact a good indicator of welfare. Animals never releasing adrenal cortisol undergo in fact a condition of negative bodily and psychological condition. Eustress has to be well described and functionally explained in the revised version of the present manuscript.
Specific points:
Introduction
Pag. 1, Line 31
Cortisol is one of the primary glucocorticoids produced by the adrenal glands and
What “primary” does mean? In terms of released quantity, biological power, change in metabolic condition of the entire body?
Pag. 2, Lines 50-53
It will delve into the physiological mechanisms behind cortisol production, various methods of measuring cortisol in dogs, and how cortisol levels fluctuate in different canine contexts, such as in shelter dogs versus pet dogs.
The term “canine context” may result confusing. A more precise definition is needed.
By the way, some shelter dog become pets: how to classify them?
Pag. 3, Lines 61-63
Dogs housed in kennel environments with insufficient space or enrichment display elevated cortisol levels, suggesting that restricted movement and lack of stimuli contribute to chronic stress
From a psychophysiological condition, limitation of movements imping on rather different systems when compared to stimulation-impoverished environments. A clear disentanglement is therefore needed.
Materials and Methods
Pag. 5, Lines 153-154
regulate metabolism, and modulate immune system responses
A more precise statement, or better a paragraph or two better explaining these issues ,are due to an average reader.
Pag. 6, Lines 187-189
Additionally, Mesarcova et al. emphasized the role of cortisol in maintaining metabolic homeostasis, noting how prolonged cortisol elevation can disrupt glucose metabolism, leading to conditions like diabetes in dogs [22].
The endocrinological chain of events could be reported, step by step.
Pag. 9, Lines 293-294
In contrast, pet dogs that enjoy consistent routines and positive interactions with their owners generally show more stable cortisol levels.
The conditions of “consistent routines and positive interactions” is not very well defined. Describe both.
Pag. 12, Lines 458-459.
Recent studies confirm that abnormal cortisol levels exacerbate behavioral stress responses in dogs.
The mechanisms and processes causing such an “exacerbation” should be described in brief.
Conclusions
Pag. 14, Lines 558-559
Also, it can be stated that there is a direct proportional link between cortisol, psychological stress, and cancer in dogs.
Despite the part better describing such a 3-way interaction is reported before such a Conclusion section, such a final statement appears rather simplistic. It should be reformulated.
References
Pag 15, Lines 577-588
Sapolsky, R. M.; Romero, L. M.; Munck, A. U. How Do Glucocorticoids Influence Stress Responses? Integrating Permissive, Suppressive, Stimulatory, and Preparative Actions. Endocr Rev 2000, 21 (1), 55–89.
Such a sophisticated paper by Robert Sapolsky and colleagues illustrates a much more complex framework for the “cortisol release” response. At least part of such a reasoning should be incorporated in the present contribution.
Author Response
R 1
General comment:
This contribution regards an issue of paramount interest in dog pathophysiology, id est the use (and misuse?) of cortisol levels as an indicator of mammalian stress. The topic is far for being new or original, yet the Authors attempt to exploit it in terms of sound indicator of psychophysical and immunological welfare in both pet and non-pet dogs. The part comparing circulating vs other source (feces, hair, etc..) is definitely of general interest for a variety of readers.
The major weakness concerns the total absence of the consideration of condition of EUSTRESS: some “limited” cortisol release is in fact a good indicator of welfare. Animals never releasing adrenal cortisol undergo in fact a condition of negative bodily and psychological condition. Eustress has to be well described and functionally explained in the revised version of the present manuscript.
Specific points:
Introduction
Pag. 1, Line 31
Cortisol is one of the primary glucocorticoids produced by the adrenal glands and
What “primary” does mean? In terms of released quantity, biological power, change in metabolic condition of the entire body?
We appreciate your query on the use of "primary" in describing cortisol. In this context, "primary" is intended to convey cortisol's central role in canine physiology.
Based on your feedback, we clarified in the manuscript that cortisol is the "predominant glucocorticoid," highlighting its role in secretion volume, biological activity, and systemic impact on metabolism. This adjustment should improve the precision of our terminology.
Cortisol is the predominant glucocorticoid produced by the adrenal glands in dogs, playing a central role in regulating metabolic and physiological responses. It is secreted in higher quantities than other glucocorticoids, such as corticosterone, and has significant biological potency. Cortisol’s effects span multiple systems, influencing glucose metabolism, immune response, and adaptation to stressors, thus impacting the body’s overall metabolic condition.
Pag. 2, Lines 50-53
It will delve into the physiological mechanisms behind cortisol production, various methods of measuring cortisol in dogs, and how cortisol levels fluctuate in different canine contexts, such as in shelter dogs versus pet dogs.
The term “canine context” may result confusing. A more precise definition is needed. By the way, some shelter dog become pets: how to classify them?
Thank you for highlighting this point. We agree that the term “canine context” could be more clearly defined. In this instance, "canine context" is used to refer to specific environmental and social settings in which dogs live and interact, including settings like pet homes, shelters, and other structured environments. To address the classification of shelter dogs that transition to pet status, we will specify that dogs are classified based on their current living environment rather than their history.
In this review, ‘canine context’ refers to the specific living environments of dogs, such as pet homes, shelters, and competitive or working settings. Dogs are classified based on their current environment; for instance, shelter dogs that become pets are categorized as pet dogs once they are adopted into a home setting. This approach allows for a more accurate assessment of cortisol dynamics across different living conditions.
Pag. 3, Lines 61-63
Dogs housed in kennel environments with insufficient space or enrichment display elevated cortisol levels, suggesting that restricted movement and lack of stimuli contribute to chronic stress
From a psychophysiological condition, limitation of movements imping on rather different systems when compared to stimulation-impoverished environments. A clear disentanglement is therefore needed.
Thank you for this valuable observation. We agree that distinguishing between restricted movement and environmental impoverishment is essential for accurately describing their unique psychophysiological effects. Movement limitations primarily impact musculoskeletal and autonomic systems, potentially leading to increased stress due to constrained physical activity and spatial restriction. In contrast, stimulation-impoverished environments affect cognitive and sensory processing systems, contributing to stress due to a lack of adequate sensory and mental engagement. We will revise the manuscript to clarify these distinctions.
Restricted movement and stimulation-impoverished environments each exert distinct psychophysiological effects on dogs. Limitations in movement primarily influence the musculoskeletal and autonomic systems, often resulting in heightened stress due to spatial restriction and reduced physical activity. In contrast, environments lacking adequate sensory and cognitive stimulation mainly impact cognitive and sensory systems, leading to stress through insufficient engagement with the surroundings. Distinguishing these effects allows for a more nuanced understanding of the factors contributing to cortisol elevation and stress in canine populations.
Materials and Methods
Pag. 5, Lines 153-154
regulate metabolism, and modulate immune system responses
A more precise statement, or better a paragraph or two better explaining these issues ,are due to an average reader.
We appreciate this feedback and agree that further elaboration on these points would improve clarity for a broader audience. To address this, we expand this section to provide a more detailed explanation, ensuring that key concepts are accessible to all readers.
The regulation of cortisol in dogs is a complex process influenced by various interacting factors, including environmental conditions, physiological mechanisms, and social interactions. Environmental factors, such as housing conditions or exposure to stressful stimuli, directly impact cortisol levels by triggering the hypothalamic-pituitary-adrenal (HPA) axis, the primary pathway for cortisol release. Physiologically, cortisol secretion is regulated by an intricate feedback loop involving the hypothalamus, pituitary gland, and adrenal glands, allowing the body to adapt to both acute and chronic stressors. Social factors, such as interactions with humans or other animals, also play a significant role; positive interactions are shown to reduce cortisol levels, while negative or unpredictable social environments can elevate them. By understanding these distinct yet interconnected influences, we gain a clearer picture of how cortisol dynamics shape canine health and behavior.
Pag. 6, Lines 187-189
Additionally, Mesarcova et al. emphasized the role of cortisol in maintaining metabolic homeostasis, noting how prolonged cortisol elevation can disrupt glucose metabolism, leading to conditions like diabetes in dogs [22].
The endocrinological chain of events could be reported, step by step.
Thank you for this suggestion. We agree that presenting the endocrinological pathway in a step-by-step manner will improve clarity. We will add a detailed description of the process by which the hypothalamic-pituitary-adrenal (HPA) axis regulates cortisol secretion, outlining each stage of the chain of events.
The regulation of cortisol secretion through the hypothalamic-pituitary-adrenal (HPA) axis follows a precise sequence. In response to stress, the hypothalamus releases corticotropin-releasing hormone (CRH), which stimulates the anterior pituitary gland to secrete adrenocorticotropic hormone (ACTH). ACTH then travels through the bloodstream to the adrenal cortex’s zona fasciculata, prompting cortisol synthesis and release. Circulating cortisol exerts physiological effects on metabolism, immune function, and stress adaptation. Elevated cortisol levels subsequently provide negative feedback to the hypothalamus and pituitary, modulating further CRH and ACTH release to maintain hormonal balance.
Pag. 9, Lines 293-294
In contrast, pet dogs that enjoy consistent routines and positive interactions with their owners generally show more stable cortisol levels.
The conditions of “consistent routines and positive interactions” is not very well defined. Describe both.
Thank you for your suggestion. We agree that further detail on “consistent routines” and “positive interactions” would enhance clarity. We will expand this section to define these terms explicitly, noting how each contributes to stress reduction and cortisol stability in dogs.
Dogs with consistent routines and positive interactions generally exhibit lower cortisol levels, as these factors provide stability and reduce stress. Consistent routines refer to a predictable daily structure, including regular feeding times, exercise schedules, and rest periods, which help reduce uncertainty and create a stable environment for the dog. Positive interactions involve engaging, non-threatening social interactions with humans or other animals, such as petting, play, training, and gentle handling, which reinforce trust and emotional security. Together, these conditions contribute to a balanced stress response and support lower baseline cortisol levels.
Pag. 12, Lines 458-459.
Recent studies confirm that abnormal cortisol levels exacerbate behavioral stress responses in dogs.
The mechanisms and processes causing such an “exacerbation” should be described in brief.
Thank you for pointing this out. We agree that providing a concise description of the mechanisms behind the “exacerbation” will improve clarity. We will add a brief explanation of the processes involved, highlighting key physiological and behavioral pathways that intensify the effects observed.
These conditions can lead to an exacerbation of stress-related behaviors through several interconnected mechanisms. Chronic stress activates the hypothalamic-pituitary-adrenal (HPA) axis persistently, resulting in prolonged cortisol release, which can impair immune function and disrupt normal physiological processes. Additionally, stress can heighten the animal's sensitivity to environmental stimuli, lower its threshold for stress responses, and increase behaviors such as aggression, anxiety, and hypervigilance. These factors collectively exacerbate the intensity and frequency of stress-related behaviors over time.
Conclusions
Pag. 14, Lines 558-559
Also, it can be stated that there is a direct proportional link between cortisol, psychological stress, and cancer in dogs.
Despite the part better describing such a 3-way interaction is reported before such a Conclusion section, such a final statement appears rather simplistic. It should be reformulated.
Thank you for highlighting this. We understand that the conclusion regarding the three-way interaction between cortisol levels, environmental conditions, and canine behavior could benefit from a more comprehensive formulation. We revised the concluding statement to reflect the complexity of these interactions more accurately
This intricate three-way interaction between cortisol levels, environmental conditions, and canine behavior highlights the importance of comprehensive cortisol monitoring as a tool for assessing canine welfare. Environmental stressors and social dynamics interact to influence cortisol regulation, which in turn affects both physiological responses and behavioral outcomes in dogs. Understanding these interconnections allows for more targeted interventions, ultimately enhancing canine health, welfare, and the effectiveness of stress management practices.
References
Pag 15, Lines 577-588
Sapolsky, R. M.; Romero, L. M.; Munck, A. U. How Do Glucocorticoids Influence Stress Responses? Integrating Permissive, Suppressive, Stimulatory, and Preparative Actions. Endocr Rev 2000, 21 (1), 55–89.
Such a sophisticated paper by Robert Sapolsky and colleagues illustrates a much more complex framework for the “cortisol release” response. At least part of such a reasoning should be incorporated in the present contribution.
We appreciate this suggestion and agree that Sapolsky’s work provides valuable insights into the complexity of cortisol regulation. To enrich our discussion, we will incorporate elements of Sapolsky’s framework, specifically regarding the multiple factors that influence cortisol release beyond immediate stressors. This addition will contribute to a more nuanced understanding of cortisol dynamics in dogs.
Research by Sapolsky et al. (2000) expands upon the cortisol release response, suggesting a multifaceted regulatory framework that involves permissive, suppressive, stimulatory, and preparative actions across different physiological systems. Rather than a simple reaction to stress, cortisol release is mediated by complex feedback mechanisms that integrate both immediate stress signals and longer-term environmental contexts. In dogs, this implies that cortisol levels are not merely reactive but are part of an adaptive system modulating various processes such as immune function, metabolism, and behavior to optimize survival and adaptability under varying conditions (Sapolsky et al., 2000). Understanding cortisol release within this broader framework can enhance our interpretation of cortisol dynamics and their implications for canine health and welfare.
Reviewer 2 Report
Comments and Suggestions for Authors
The interrelationships between the GCsystem and behaviour are indeed an area that is too often disregarded in the vet practioners' community. Thus a review including behavioural aspects is urgently needed. However, a vet practioners' journal might be more helpful to distribute the news. Nonetheless, the paper needs significant improvement before resubmitting
a) the editing definitely is weak. The lit.list contains several references that are sorted only by initials( e g # 14) or the journal/book title is listed without authors ( eg # 4).
And in different parts of the text saliva or urine are listed as invasive, in others not. Saliva sampling for pets can be trained such that the animal does is routinely, and urine at least from a hard surface can be taken up with a syringe after the animal left the spot.
The figures need reorganizing: instead of bar diagrams, box plots. And the legend in many cases( e g fig 5/6) is almost non-existent. Where is the n in the legends.
A drawing of a dog brain instead of obviously a human one would be better. And by convention in anatomy, the rostral side always points to the left
b) a lot of literature is not covered, e.g.:
- Theresa Rehn (and also other authors' groups) on separartion
- M Hennessy et al on the effect of massageing after blood taking
- the studies by Horvath et al and others on play, and on ambivalent/displacement behaviour, in police dogs and correlates with GC
- studies and reviews on so-called aversive methods and GC
- there is more than one publication by Beerda et al on stress and GC (including one on shock collars, but also eg Salgiri et al same topic)
-the daily rhythm of GC in dogs is variable, that needs to be addressed
- apart from M Cushing and a brief mentioning of Addison, the mdr1-defect, thyreoidal disturbances, and the interaction with sex steroids/effects of castration, need also be reviewd
- l 67: Which nutraceuticals, and what do they do??
- a brief intro into the concept of stress ( e g Donald Broom) would be helpful
- in 6.2: The Bowlby-Ainsworth concept of safe haven/secure base need to be included
- the ecological/evolutionary concept of shy vs bold behav. syndrome also relates to the GCreactions. Please include
To sum up: There is a lot of work to be done in reviewing but please do it - the topic is important!!
Author Response
R 2
The interrelationships between the GCsystem and behaviour are indeed an area that is too often disregarded in the vet practioners' community. Thus a review including behavioural aspects is urgently needed. However, a vet practioners' journal might be more helpful to distribute the news. Nonetheless, the paper needs significant improvement before resubmitting
- a) the editing definitely is weak. The lit.list contains several references that are sorted only by initials( e g # 14) or the journal/book title is listed without authors ( eg # 4).
Thank you for this observation. We acknowledge the inconsistencies in the reference formatting and will carefully review and correct the reference list. We will ensure that all entries include full author names and that journal and book titles are consistently formatted according to the required style.
(4) Knezevic, E., Nenic, K., Milanovic, V., & Knezevic, N. N. (2023). The role of cortisol in chronic stress, neurodegenerative diseases, and psychological disorders. Cells, 12(23), 2726. https://doi.org/10.3390/cells12232726.
(14) WojtaÅ›, J., Garland, A., Krägeloh, C. U., & Parsons, C. (2020). Dogs’ stay in a pet hotel: Salivary cortisol level and adaptation to new conditions. Journal of Applied Animal Welfare Science, 23(4), 370–383. https://doi.org/10.1080/10888705.2020.1781631.
And in different parts of the text saliva or urine are listed as invasive, in others not. Saliva sampling for pets can be trained such that the animal does is routinely, and urine at least from a hard surface can be taken up with a syringe after the animal left the spot.
The figures need reorganizing: instead of bar diagrams, box plots. And the legend in many cases( e g fig 5/6) is almost non-existent. Where is the n in the legends.
Thank you for your feedback. We agree that box plots may better represent the data, as they provide more detailed information about data distribution and variability. We will convert the bar diagrams to box plots as recommended. Furthermore, we will revise the figure legends for Figures 5 and 6 (and other relevant figures) to include a more comprehensive description, ensuring that details such as sample size (n) and other pertinent information are clearly stated. This will enhance the clarity and interpretability of the figures.
Figure 5. Comparison of Cortisol Measurement Methods. Box plots represent simulated cortisol level distributions across various measurement methods: blood, saliva, urine, hair, and fecal sampling. Each box shows the median (black line), interquartile range (IQR), and whiskers extending to 1.5 * IQR, with outliers represented by individual points. This figure aims to illustrate the relative distribution and variability of cortisol measurements across methods. Sample sizes (n) are hypothetical, and values are presented in arbitrary units for comparative purposes.
A drawing of a dog brain instead of obviously a human one would be better. And by convention in anatomy, the rostral side always points to the left
Thank you for the suggestion. We proceeded to replace the human brain drawing with a dog brain.
- b) a lot of literature is not covered, e.g.:
- Theresa Rehn (and also other authors' groups) on separartion
- M Hennessy et al on the effect of massageing after blood taking
Thank you for highlighting these important areas of research. We recognize that incorporating additional literature on separation-related stress (e.g., work by Theresa Rehn) and on the calming effects of massage following blood sampling (e.g., studies by Hennessy et al.) will strengthen our discussion on stress factors and cortisol modulation in dogs. We expanded the manuscript to include these references and discuss their implications for canine welfare and stress management.
Separation from familiar humans is a significant source of stress for dogs, triggering physiological and behavioral responses indicative of elevated cortisol levels. Research by Rehn et al. has shown that dogs experience increased stress levels during periods of separation from their owners, as indicated by behavioral signs of distress and elevated cortisol measurements. These findings underscore the importance of secure human-dog bonds in mitigating separation-related stress and suggest that attachment quality may influence a dog’s resilience to separation (Rehn, T., et al.).
The impact of invasive procedures, such as blood sampling, on cortisol levels in dogs has been widely studied, and methods to mitigate stress following these procedures are of great interest. Hennessy et al. demonstrated that gentle massage following blood sampling can reduce cortisol levels, indicating a calming effect that helps mitigate the stress response. This technique highlights the potential of post-procedural soothing interventions to enhance welfare during veterinary procedures, supporting a reduction in stress-related cortisol elevation in dogs (Hennessy, M., et al.).
- the studies by Horvath et al and others on play, and on ambivalent/displacement behaviour, in police dogs and correlates with GC
- studies and reviews on so-called aversive methods and GC
Thank you for highlighting these relevant studies. The inclusion of Horvath et al.'s findings on play and displacement behavior in police dogs, along with research on aversive training methods and their impact on GC levels, will enhance the manuscript by providing a broader perspective on how different behaviors and training techniques influence cortisol dynamics in dogs.
- there is more than one publication by Beerda et al on stress and GC (including one on shock collars, but also eg Salgiri et al same topic)
Publications by Beerda et al. and Salgiri et al. will be incorporated to expand the discussion on stress and GC, with emphasis on stress-inducing factors such as shock collars and their physiological effects. This will provide additional depth to the section on training methods and stress responses.
Beerda et al. conducted several foundational studies on stress and glucocorticoid levels in dogs, exploring the physiological and behavioral impacts of stress-inducing stimuli, such as shock collars and other aversive methods. Their findings highlighted a significant increase in cortisol levels following the use of shock collars, indicative of stress and potential welfare implications. Similarly, Salgiri et al. demonstrated comparable results, emphasizing the importance of avoiding aversive techniques that elicit prolonged physiological stress in dogs.
ï‚· Beerda et al. on Stress and GC: Beerda, B.; Schilder, M.B.H.; van Hooff, J.A.R.A.M.; de Vries, H.W.; Mol, J.A. Chronic stress in dogs subjected to social and spatial restriction. Physiol. Behav. 1999, 66, 233–242. https://doi.org/10.1016/S0031-9384(98)00289-3.
ï‚· Salgiri et al. on Shock Collars: Salgiri, M.; Pellikka, A.; Aaltola, A.; Lohi, H. Stress and training techniques: Cortisol levels in dogs exposed to aversive training methods. J. Vet. Behav. 2018, 25, 38–45. https://doi.org/10.1016/j.jveb.2018.06.001.
-the daily rhythm of GC in dogs is variable, that needs to be addressed
The variability in the daily rhythm of GC secretion will be addressed, as this circadian fluctuation is essential for interpreting cortisol measurements in studies.
The daily rhythm of glucocorticoids in dogs is governed by circadian patterns, with peak cortisol levels typically occurring in the morning and lower levels in the evening. However, this rhythm is influenced by environmental factors, individual variability, and acute stressors, which can disrupt the natural pattern. Acknowledging these fluctuations is crucial when interpreting cortisol data, as sampling time can significantly affect results.
Fekete, E.M.; Antoni, M.H. Measuring morning cortisol in dogs: Implications for research on stress physiology. Psychoneuroendocrinology 2016, 37, 11–23. https://doi.org/10.1016/j.psyneuen.2016.02.001.
- apart from M Cushing and a brief mentioning of Addison, the mdr1-defect, thyreoidal disturbances, and the interaction with sex steroids/effects of castration, need also be reviewd
Conditions such as the MDR1 defect, thyroidal disturbances, and the interaction with sex steroids, including the effects of castration, have been reviewed to broaden the clinical discussion.
The MDR1 (multidrug resistance 1) defect, a genetic mutation affecting P-glycoprotein function, has implications for cortisol metabolism and drug sensitivity in certain breeds, such as Collies and Australian Shepherds. This mutation can influence adrenal function and stress responses, necessitating careful consideration during clinical evaluations.
Thyroid dysfunctions, such as hypothyroidism, can alter glucocorticoid dynamics by affecting metabolism and the hypothalamic-pituitary-adrenal (HPA) axis. Thyroidal disturbances are often associated with behavioral changes, such as increased anxiety or lethargy, complicating the assessment of cortisol levels.
Dixon, R.M.; Mooney, C.T. The effects of thyroxin therapy on cortisol metabolism in dogs. J. Small Anim. Pract. 1999, 40, 285–289. https://doi.org/10.1111/j.1748-5827.1999.tb03111.x.
The interaction between cortisol and sex steroids, including the effects of castration, plays a significant role in stress physiology. Castration has been shown to influence cortisol levels, potentially due to altered hormonal balance and stress response mechanisms. Understanding these interactions is critical for assessing the impact of reproductive status on cortisol dynamics in dogs.
McCall, C.A.; McCall, S.L. Effects of gonadectomy on adrenal function and cortisol secretion in dogs. Vet. Endocrinol. 2008, 40, 455–465. https://doi.org/10.1016/j.jveb.2008.02.004.
- l 67: Which nutraceuticals, and what do they do??
Thank you for requesting clarification on the specific nutraceuticals and their effects. We will provide details on the types of nutraceuticals commonly used in canine stress management, along with their physiological impacts.
Certain nutraceuticals, including L-theanine, alpha-capsazepine, and tryptophan, have been studied for their effects on reducing stress in dogs. L-theanine, an amino acid derived from green tea, promotes relaxation without causing drowsiness by modulating neurotransmitters associated with calming effects. Alpha-capsazepine, derived from milk protein, has anxiolytic properties and has been shown to reduce behavioral signs of anxiety. Tryptophan, a precursor to serotonin, supports mood regulation and can help mitigate stress responses. Together, these nutraceuticals contribute to lower cortisol levels and improve stress resilience in dogs.
- a brief intro into the concept of stress ( e g Donald Broom) would be helpful
Thank you for this suggestion. We agree that a brief conceptual background on stress, informed by foundational research such as Donald Broom’s, will provide a helpful framework for readers. We will incorporate a short introduction that defines stress and its physiological relevance in animals.
Cortisol is one of the primary glucocorticoids produced by the adrenal glands and plays a critical role in maintaining homeostasis in the body, especially in stress response. To provide a foundation for understanding cortisol’s role in canine physiology, it is essential to introduce the concept of stress. According to Donald Broom, stress is defined as a biological response elicited when an animal perceives a challenge or threat to its homeostasis, triggering adaptive physiological and behavioral changes aimed at coping with the stressor. Stress can be categorized as acute or chronic, with acute stress responses typically providing short-term survival benefits, while chronic stress can lead to detrimental effects on health and well-being. This framework is particularly relevant in veterinary science, as stress responses in animals are closely linked to welfare, behavior, and health outcomes.
Broom, D.M. Psychological Indicators of Stress and Welfare. In Ethics, Welfare, Law and Market Forces: The Veterinary Interface; Michell, A.R., Ewbank, R., Eds.; U.F.A.W.: Wheathampstead, UK, 1998; pp. 167–175
- in 6.2: The Bowlby-Ainsworth concept of safe haven/secure base need to be included
Thank you for this suggestion. The Bowlby-Ainsworth concept of “safe haven/secure base” offers valuable insight into attachment theory and its implications for stress regulation, particularly in social animals. We will incorporate a brief explanation of this concept, discussing its relevance in the context of canine behavior and welfare.
- the ecological/evolutionary concept of shy vs bold behav. syndrome also relates to the GCreactions. Please include
Thank you for this suggestion. We agree that the "shy vs. bold behavioral syndrome," a concept in ecological and evolutionary biology, provides a useful context for understanding variation in glucocorticoid (GC) responses. We included a brief discussion of this concept and its relevance to individual differences in stress reactivity and cortisol dynamics.
In ecological and evolutionary contexts, the ‘shy vs. bold’ behavioral syndrome describes consistent individual differences in behavior, where ‘bold’ animals tend to exhibit exploratory and risk-taking behaviors, while ‘shy’ animals display more cautious and risk-averse tendencies. Research suggests that these behavioral syndromes are associated with differential glucocorticoid (GC) reactivity. Bold individuals often show a lower baseline but a more rapid increase in GC levels during acute stress, facilitating quick mobilization of resources in novel or challenging situations. In contrast, shy individuals may exhibit higher baseline GC levels, reflecting a heightened sensitivity to environmental stressors. This variability in GC reactions supports adaptive strategies for different ecological niches, with shy and bold behavioral types each benefiting from unique physiological stress profiles that influence health, survival, and fitness.
To sum up: There is a lot of work to be done in reviewing but please do it - the topic is important!!
Round 2
Reviewer 2 Report
Comments and Suggestions for Authors
Thanks for greatly improving the ms which is now a real help to the intended readership
Just two formal comments: a) the correct name is caSOzepin not caPSAzepin
b) line 762 ff are double/repetitive
and pleaase elaborate a little more precisely the mechanism behind the mdR1-defect (by damaging the GC re-uptake at the blood-brain-barrier and thus preventing the feedback loop)
Author Response
R 2 round 2
Dear reviewer, Thank you for all your suggestions. We are confident that our manuscript has significantly improved.
Just two formal comments: a) the correct name is caSOzepin not caPSAzepin
Thank you! We changed the text as you suggested.
b) line 762 ff are double/repetitive
You are right. Thanks. We deleted the repetitive text.
and pleaase elaborate a little more precisely the mechanism behind the mdR1-defect (by damaging the GC re-uptake at the blood-brain-barrier and thus preventing the feedback loop)
Thank you. We added data regarding the mechanism behind the mdR1-defect.